# The Prospective Effects of Coping Strategies on Mental Health and Resilience at Five Months after HSCT

**DOI:** 10.3390/healthcare11131975

**Published:** 2023-07-07

**Authors:** Maya Corman, Michael Dambrun, Marie-Thérèse Rubio, Aurélie Cabrespine, Isabelle Brindel, Jacques-Olivier Bay, Régis Peffault de La Tour

**Affiliations:** 1LAPSCO UMR CNRS 6024, Université Clermont Auvergne (UCA), 34 Avenue Carnot, 63000 Clermont-Ferrand, France; 2Service D’Hématologie, CHRU Nancy-Hôpitaux de Brabois, 54511 Vandoeuvre-les-Nancy, France; 3CHU de Clermont-Ferrand, Site Estaing, Service de Thérapie Cellulaire et D’hématologie Clinique Adulte, 63000 Clermont-Ferrand, France; 4Service D’Hématologie, Hôpital Saint-Louis, Greffe de Moelle, 75010 Paris, France

**Keywords:** hematopoietic stem cell transplant (HSCT), adjustment coping strategies, mental health, quality of life, post-traumatic stress disorder, resilience

## Abstract

Objectives: Hematopoietic stem cell transplantation (HSCT) is a stressful event that engenders psychological distress. This study examines the prospective effects of coping strategies during hospitalization on resilience and on various mental-health dimensions at five months after transplantation. Methods. One hundred and seventy patients (M*_age_* = 52.24, *SD* = 13.25) completed a questionnaire assessing adjustment strategies during hospitalization, and 91 filled out a questionnaire five months after HSCT (M*_age_* = 51.61, *SD* = 12.93). Results: Multiple regression analyses showed that a fighting spirit strategy positively predicted resilience (*p* < 0.05), whereas anxious preoccupations predicted anxiety (*p* < 0.05), poorer mental QoL (*p* < 0.01), and were associated with an increased risk of developing PTSD (OR = 3.27, *p* < 0.01; 95% CI: 1.36, 7.84) at five months after transplantation. Hopelessness, avoidance, and denial coping strategies were not predictive of any of the mental health outcomes. Finally, the number of transplantations was negatively related to a fighting spirit (*p* < 0.01) and positively related to hopelessness-helplessness (*p* < 0.001): Conclusions: These results highlight the importance of developing psychological interventions focused on coping to alleviate the negative psychological consequences of HSCT.

## 1. Background

Hematopoietic Stem Cell Transplantation (HSCT) can significantly impair several psychological health aspects of the individuals who benefit from this treatment. It alters short- and long-term physical and mental Quality of Life (QoL) [1,2,3] and engenders symptoms of psychological distress, such as anxiety [1,4], depression [1,5], and post-traumatic stress disorder (PTSD) [1,6]. The pre-graft period is a source of psychological distress because of both the treatment and the anticipation of hospitalization [7]. Hospitalization, particularly isolation in a protective area, is recognized as distressing for patients [8,9,10,11,12,13]. Regarding these issues, studies are necessary to explore the psychological factors involved in the nature and quality of the psychological and physical health of patients during the early stage of hospitalization and its impact several months later.

This question can be addressed through the exploration of coping strategies used during HSCT hospitalization. These coping strategies include behavioral, cognitive, and emotional responses used to reduce or manage psychological distress engendered by a stressful and threatening situation that exceeds the patient’s external and/or internal personal resources [14]. According to the transactional model [14], the coping process is dynamic and depends on the individual’s perception and assessment of his/her relationship to his/her environment in terms of perceived stress, control, and social support. Generally, studies in oncology show that the perception of good social support [15,16], a situation perceived as controllable [17], and lower or moderate perceived stress [18,19] are associated with active coping and better mental health and QoL. There are no bad or good coping strategies per se, but some are considered to be linked to better mental health and QoL than others [20,21]. Therefore, coping strategies can be conceptualized as adaptive (e.g., problem solving, positive reappraisal) or maladaptive (e.g., negative rumination or emotional suppression) and as two independent processes that can interact [21,22]. More precisely, adaptive coping seems to mediate the relationship between positive psychological resources or dispositions (e.g., resilience, optimism) and mental health in the sense of protecting and buffering psychological distress in the short and long term [21,23,24]. Conversely, negative psychological dispositions, such as neurotic personality traits (i.e., the tendency to experience negative emotions such as sadness, anxiety, nervousness, and tension), and both mental health and QoL impairment are associated with maladaptive coping strategies [25].

Studies testing the prospective effects of coping strategies on mental health and resilience after HSCT are scarce. In the case of hematological malignancies, a meta-analysis [26] showed that a fighting spirit (i.e., an active and engaged coping strategy) is positively linked with a better QoL [27], whereas hopelessness-helplessness, fatalism, or anxious preoccupations (i.e., passive and disengagement strategies) are positively related to psychological distress [28]. In HSCT, to our knowledge, four prospective studies examined the relationships between depression and coping [29,30,31,32]. They mainly reveal that the greater the depression symptomatology, the more often an individual will use maladaptive coping strategies such as avoidance or fatalism, and that such an individual will rarely use adaptive coping strategies (i.e., fighting spirit, problem solving). The results of this study revealed that high scores of depression symptomatology are associated with a more frequent use of maladaptive coping strategies such as avoidance or fatalism and a less frequent use of adaptive strategies such as a fighting spirit or problem-solving. Finally, lower social support and higher use of an avoidance coping strategy one month prior to transplantation predicted greater PTSD symptomatology seven months after transplantation [33].

Our study explored the prospective effects of five coping strategies during HSCT on several markers of mental health (i.e., QoL, depression, anxiety, happiness, and PTSD symptomatology) and on resilience in a five-month follow-up. First, we examined the relationships between perceived stress, control, perceived social support and adjustment coping strategies with a sample of people undergoing HSTC. The objective was to verify that our measures correlated with each other in the expected direction, in accordance with the existing literature, thus attesting to the validity of our scales. Second, we tested the prospective effects of coping strategies on various dimensions of mental health and resilience.

In sum, hospitalization in a protected area is relatively specific to the case of bone marrow transplantation (isolation, risk of infection, aplasia, side effects of treatment during transplantation, etc.). This stage generates stress, which in turn generates specific coping strategies. We know that coping strategies partly explain certain aspects of mental health, such as anxiety, depression, post-traumatic stress symptoms and quality of life. The main objective of this research was to identify the coping strategies that predict mental health and resilience five months after HSCT.

The research will enable us to identify the most suitable strategies midway through the HSCT process, so that we can implement more targeted and effective psychotherapeutic interventions at a later stage.

We predicted that maladaptive coping strategies during hospitalization (i.e., avoidance, hopelessness-helplessness, anxious preoccupations, and denial) would positively predict PTSD, anxiety, and depression and negatively predict QoL, happiness, and resilience at five months after transplantation. We anticipated the opposite for the adaptive coping strategies (i.e., fighting spirit). The prospective effects of coping strategies on several medical outcomes (e.g., acute graft versus host disease [GvHD], relapse, death) were also explored.

## 2. Methods

### 2.1. Participants

Two hundred and fifty-seven participants were invited to participate in the “Psy-Greffe” protocol between November 2017 and September 2020. Among them, 70 declined to participate or could not participate for various reasons. The recruited sample filled out three questionnaires: one before hospitalization, one during transplantation, and one five months after the allograft. One hundred and eighty-seven participants filled out the first questionnaire (M*_age_* = 52.07, *SD* = 13.22, age range from 19 to 72 years old) and 170 filled out the second one (M*_age_* = 52.24, *SD* = 13.25, age range from 19 to 72 years old). Finally, 91 completed the third questionnaire at the five-month follow-up (M*_age_* = 51.61, *SD* = 12.93, age range from 23 to 70 years old). They came from three hospital centers in France and were candidates for an allogenic Hematopoietic Stem Cell Transplantation (HSCT) after diagnoses of hematologic malignancies. (See Figure 1). We estimated the required sample size for sufficient correlation power (90%). On the basis of the correlations between coping strategies and QoL reported by O’Connor et al. [28] (i.e., in absolute value, r between 0.34 and 0.67), the minimum required sample size was 87 with *r* = 0.34.

### 2.2. Procedures

The study population concerned all patients proposed for an allograft, aged over 18 years, who did not object to answering the various questionnaires over a period of approximately seven months post-transplant. The participants came from the hematology departments in Paris, Clermont-Ferrand, and Nancy. During the pre-graft interview with the doctor, the study was proposed to each eligible patient by reading an information note about the protocol. After a 15-day cooling-off period, patients who agreed to take part in the study filled out informed consent forms and provided sociodemographic and medical information. Their levels of QoL, anxiety, depression, and happiness were also assessed at this time (Time 0). Next, a questionnaire assessing coping, perceived control, stress, and social support was given during hospitalization, between day one and day seven after transplantation (Time 1). The third questionnaire was proposed five months (+/− one month) after the allograft (Time 2). This final questionnaire measured QoL, anxiety, depression, happiness, PTSD, and resilience. The relevant medical data were extracted from the ProMISe (Project Manager Internet Server) database. Participants knew they were not retributed for their participation. All participants were volunteers.

### 2.3. Measures

**Adjustment strategies** (Time 1)**.** Mental adjustment to cancer scale (MACs) [34,35]. This French version contains 45 items measuring five different coping strategies used by patients with a cancer diagnosis. This is a 4-point Likert-type scale (from 1 “not at all” to 4 “completely”). The scale is divided into five subscales for each meta-coping strategy: fighting spirit (*α* = 0.81), hopelessness-helplessness (*α* = 0.80), anxious preoccupations (*α* = 0.89), avoidance (*α* = 0.75) and denial (*α* = 0.80). To obtain a global score of maladaptive adjustment strategies, we summed up the four means of hopelessness-helplessness, anxious preoccupations, avoidance, and denial scores. Some statements were adapted to match with the HSCT conditions of patients.

**Perceived social support** (Time 1)**.** Social Provisions Scale-Revised (SPS-R) [36,37]. Twenty-four items measure the perceived social support on a four-point scale from 1 “strongly disagree” to 4 “strongly agree.” Six subscales assess the different dimensions of social support: emotional support (attachment), social integration, reassurance of value (reassurance of worth), material assistance (reliable alliance), advice and information (guidance), and the need to feel useful (opportunity for nurturance). The total score is the sum of all subscale scores and varies from 0 to 96 (*α* = 0.84). The higher the score, the more people perceive good social support.

**Perceived stress** (Time 1). The Perceived Stress Scale (PSS) [38,39] assesses perceived stress regarding 10 items on a five-point scale from 0 “never” to 4 “very often.” The total score is the sum of all items. A score between 0 and 13 means low stress; a score between 14 and 26 indicates medium stress, and a score between 27 and 40 means that people report a high level of perceived stress. The internal validity is very satisfying (*α* = 0.87).

**Perceived control** (Time 1). Internal, Powerful others, and Chance Locus of Control (IPC) [40]. This is a Canadian version of Levenson’s (1973) [41] three-dimensional locus of control assessment of 24 items. This is a 6-point Likert-type scale (from 1 “totally agree” to 6 “totally disagree”). Eight items measure the “Internality” dimension (*α* = 0.64), eight the “Powerful others” dimension (externality), and eight the “Chance” dimension (externality). We added both “Powerful others” and “Chance” dimensions to obtain an externality score (*α* = 0.77).

**Quality of Life** (Time 0 and Time 2). We used the SF-12 [42], a short version of the SF-36 which assessed both mental/social (i.e., vitality, social functioning, role-emotional, mental health) and physical (physical functioning, role-physical, bodily pain, general health) QoL. Calculation of Cronbach’s alpha is not possible because all items are weighted. Each item score is converted into standardized data.

**Mental Health/Illness** (Time 0 and Time 2). Anxiety and Depression symptomatology was measured with the Hospital Anxiety and Depression scale (HAD) [43]. Seven items estimate the anxiety symptomatology (*α*_t0_ = 0.76, *α*_t2_ = 0.72) and seven items assess symptoms of depression (*α*_t0_ = 0.70, *α*_t2_ = 0.80). Happiness was assessed with the Subjective Authentic-Durable Happiness scale (SA-DHS) [44] through 13 items (*α*_t0_ = 0.95, *α*_t2_ = 0.97).

**Post-Traumatic Stress Disorder** (Time 2). Post-Traumatic Stress Disorder Checklist Scale (PCLS) [45,46]. This scale is used to detect post-traumatic stress disorder through 17 items assessing the severity of 17 symptoms of PTSD listed in the DSM-V. For each item, individuals indicate how much they have experienced these symptoms during the last month from 1 (“Not at all”) to 5 (“Very often”). This scale has a very good internal consistency (*α* = 0.91).

**Resilience** (Time 2). The Connor-Davidson Resilience scale (CD-RISC) [47,48] comprises 25 items, each rated on a 5-point scale (from 0 “not at all” to 4 “almost all the time”), with higher scores reflecting greater resilience. The internal consistency in our sample is very satisfying (*α* = 0.87).

**Medical and socio-demographic variables.** Patients provided some information about their sex, age, educational level, marital status, and socio-professional category. Controlled medical variables included alcohol consumption, smoking, physical activity, body mass index (BMI), sleeping hours, type of disease, number of transplantations, the latency time between disease diagnosis and transplantation, myeloablative conditioning, the type of donor and chronic GvHD. Whether or not there was acute GvHD, latency between transplantation and engraftment, a relapse, a number of infections, or death during/following the hospitalization were included as dependent variables in our study with a mean follow-up of seven months (*SD* = 3.96).

**Statistical Analyses**. Data were expressed in numbers and percentages for categorical variables and as mean ± standard deviation (*SD*) for quantitative variables. Statistics were computed using Jamovi 2.3.24.0 and in this paper, *p*-value was chosen as an indicator to compare significant values through all tests used [49]. In a first step, we examined the relationships between the various variables using Pearson correlations. Then, in a second step, prospective effects were investigated using multiple linear regression (MLR) analyses. In these analyses, we calculated the z-score for all independent variables as well as for the covariates. For each MLR, we entered the different coping strategies that were significantly correlated with the selected health outcome (in step 1) as independent variables and the same health outcome as the dependent variable. Thus, an MLR analysis was realized with each health outcome (i.e., QoL, anxiety, depression, happiness, resilience, and PTSD), resulting in a series of six multiple regression analyses. Whenever available, we included the health outcome at time 0 as a covariate. Taking QoL as an example, our independent variables were hopelessness at time 1, anxious preoccupations at time 1 and quality of life at time 0. Our dependent variable was quality of life at time 2. None of the socio-demographic and medically controlled variables correlated significantly with our main dependent variables (i.e., QoL, anxiety, depression, happiness, resilience and PTSD) except the socio-professional category for anxiety, and gender for happiness. Consequently, we included the socio-professional category as a covariate in the MLR analysis with anxiety as the dependent variable, and we included participants’ gender as a covariate in the MLR analysis with happiness as the dependent variable. Multicollinearity was adequate (i.e., all *VIF*s < 3). Homoscedasticity was checked using a scatterplot with residuals against the dependent variable. No extreme values influenced the results of the multiple regression analyses (all Cook’s distance < 0.025). Except for depression as a DV (*p* < 0.01), the normality tests did not reject the normality hypothesis (all *p* > 0.10). A square root transformation was applied to the measure assessing depression at T2. As this transformation recovered normality (*p* > 0.43), we used it as the dependent variable in the correlation and multiple regression analyses. In all analyses, we used all available data without any imputations for missing data.

## 3. Results

### 3.1. Descriptive Statistics

The characteristics of our sample are available in Table 1. Among the patients in the sample who provided information, 42.7% were women, 46.4% were married, 46.3% had graduated, and 69.6% were employed. Thirty-six percent were allogenic HSCT candidates for acute leukemias, 17.4% for myelodysplastic syndromes, 10.1% for myeloproliferative neoplasia, and 11.8% for non-Hodgkin’s lymphomas. For 38% of patients, the graft came from a matched unrelated donor.

Perceived stress in our sample (*M* = 14.25, *SD* = 6.8) was moderate because it ranged between 14 and 26 on scale (The mean for PTSD symptomatology in our sample was 31.26 (*SD* = 11.86), with 16.3% of patients meeting the criteria for PTSD (i.e., a score above or equal to 44) and 32.6% meeting the criteria for psychological distress regardless of their post-traumatic condition (i.e., score above 34). In order to identify how our sample compared to the norm (through a representative sample),a t-test was used to compare the quality of life score of our sample at time 2 with an average score of the general population in France (*n* = 2743) obtained from an SF-12 validation study. Results showed that the score for quality of life was significantly lower in the sample than the average score of the general French population [42] (mental QoL: *M* = 45.60 instead of 51.2, *SD* = 8.47, t = −3.15, *p* < 0.01, *M*_diff =_ −2.79; physical QoL: *M* = 41.04 instead of 48.4, *SD* = 8.52, *t* = −11.38, *p* < 0.001, *M*_diff_ = −10.16). In the sample, 59.3% had a score below the mean of the general population for the mental component of QoL, whereas 87.9% had a level of physical QoL below the mean of a non-clinical sample. Among the patients, 19.1% had a symptomatology of depression (i.e., a score above 7; 21.1% of women and 17.6% of men) and 37.1% of patients (44.7% of women and 31.4% of men) had a score above the threshold level for clinical anxiety (i.e., a score above 7).

### 3.2. Relationships between Perceived Stress, Perceived Social Support, Perceived Control, and Adjustment Coping Strategies at t1

We used the correlation analyses between all measures at t1 to verify accordance with the transactional model of stress in the case of HSCT. Pearson correlations demonstrated a significant relationship between perceived social support and all five coping strategies (see Table 2). Greater social support was associated with a higher fighting spirit and with lower maladaptive strategies. Anxious preoccupations, hopelessness-helplessness, and a low fighting spirit were negatively and significantly related to perceived stress but not avoidance and denial. Finally, while only fighting spirit was positively related to internality, externality was positively associated with all four maladaptive strategies (i.e., hopelessness-helplessness, avoidance, denial, and anxious preoccupations).

### 3.3. The Prospective Effects of Coping Strategies on Quality of Life, Mental Health, PTSD, and Resilience at Five Months after Transplantation

Table 3 presents the Pearson correlations between coping strategies and health outcomes (see the column zero-ordered effects; i.e., “Z-O”). To examine the prospective effect of coping strategies, a series of multiple regression analyses was performed (see Table 3; columns “Adjusted *β*”).

Concerning mental QoL (as a DV), adjusting for mental QoL at Time 0 (i.e., prior to hospitalization), only the anxious preoccupations coping strategy (*β =* −0.42, *p* < 0.01) significantly and negatively predicted QoL at Time 2 (at 5-month follow-up).

Concerning anxiety as a DV, while controlling for anxiety and the socio-professional category at Time 0, the analyses showed that the anxious preoccupations coping strategy (*β* = 0.34, *p* < 0.05) was the only robust predictor of anxiety symptomatology at the follow-up.

As shown in Table 3, depression was not significantly robustly predicted by any adjustment strategies. The results were similar with happiness. Adjusting for happiness at Time 0 and for sex, happiness at Time 2 was only marginally predicted in the expected direction by a fighting spirit (*β* = 0.19, *p* < 0.10).

Because both resilience and PTSD were not assessed at Time 0, we adjusted them for mental QoL at Time 0. Resilience was only significantly predicted by a fighting spirit (*β* = 0.33, *p* < 0.05). Finally, concerning PTSD symptomatology, the anxious preoccupations coping strategy (*β* = 0.72, *p* < 0.001) was predictive of a higher level of PTSD at five months post-transplantation. Transforming the PTSD scores into a binary variable (PTSD: 0 = no; 1 = yes), and adjusting for mental QoL at Time 0, the anxious preoccupations coping style was associated with an increased risk of developing PTSD five months after HSCT (OR = 3.27, *p* < 0.01; 95% CI: 1.36, 7.84).

Finally, the prospective effects of perceived stress, perceived control, and perceived social support on QoL, PTSD, depressive and anxious symptomatology, happiness, and resilience were also explored. A series of regression analyses revealed that adjusting for mental QoL at Time 0, only perceived stress (*β =* 0.31, *p* < 0.05) significantly predicted PTSD symptomatology at five months. However, this effect disappeared when the anxious preoccupations coping strategy was included in the model (*β =* −0.08, *p* = 0.62).

### 3.4. Relationships between Coping Strategies and Medical Outcomes

We proceeded to a bivariate correlation analysis to explore the relationships between coping strategies and medical outcomes. We found that only the number of transplantations was negatively related to a fighting spirit (*r* = −0.25, *p* < 0.01) and positively related to hopelessness-helplessness (*r* = 0.30, *p* < 0.001). The higher the number of previous transplantations, the less frequently patients used the fighting spirit coping strategy, and the more often they used the hopelessness/helplessness strategy.

## 4. Discussion

The aim of this study was to test the prospective effects of coping strategies on QoL, PTSD, depressive and anxious symptomatology, happiness and, finally, resilience at five months after HSCT. There are few studies examining these relationships in cases of threatening diseases or treatments.

First, we examined the relationships between coping and perceived stress, control, and social support. Having good social support, low perceived stress, and a low external locus of control were related to less maladaptive coping strategies and more adaptive ones. Social support is well-recognized as a protective factor for adjustment in HSCT [32]. Despite the negative effect of perceived stress on the adjustment process [18,19] in the present study, both avoidance and denial were not related to the perceived stress scale. This result may suggest that such strategies protect in the short term against the perception of being stressed, even if the long-term adverse effects of these strategies are well-known [50]. In addition, our sample had a low to moderate amount of perceived stress. It may be that a certain level of perceived stress must be reached to trigger more denial or avoidance coping strategies. It would also be relevant to separately consider each adaptive/maladaptive coping strategy according to their function at a specific step or condition of the disease [51].

Finally, during HSCT, a lack of control was related to maladaptive coping strategies. The effects of internality and externality is not clear in the literature. Perceived internal control is linked to a fighting spirit [23], but internal control can increase the patient’s feeling of responsibility for the disease process and thus be deleterious in terms of health [23,52]. Interestingly, the higher the number of transplantations, the more frequently patients used the hopelessness-helplessness strategy and the less often they used the fighting spirit coping strategy. The fact that some patients had already been through this event seemed to increase this maladaptive coping, which highlights that multiple hospitalizations for HSCT represent a challenge for the quality of adaptation.

The results concerning the prospective effects of coping strategies reveal several interesting findings. First, avoidance strategy was not related to any of the mental health outcomes. It is not consistent with other studies highlighting the relationship between avoidance coping and psychological distress [29,32] or PTSD [33].

To our knowledge, no study has put forward the predictive effect of coping on QoL five months post-transplantation. The main factors studied, and involved in impairment of QoL, after auto- and allo-HSCT are medical (i.e., GvHD, regimen conditioning); socio-demographic (being young and female); environmental (lack of social support); and psychological (pre-existing psychological distress) [53]. Anxiety preoccupation was the only strategy we found to be significantly and negatively related to QoL at follow-up. Targeting factors such as perceived stress or anxiety, which are related to this maladaptive strategy, would reduce the deleterious effects of these coping strategies on QoL several months after hospitalization.

Interestingly, a fighting spirit was only related to resilience, a factor characterizing a positive recovery. For example, the degree of resilience seems to be a relevant factor that distinguishes patients who coped better with post-HSCT. Indeed, patients with high levels of resilience reported higher levels of QoL and lower anxiety and depression symptoms [54]. These results on the effect of a fighting spirit suggest that other factors potentially involved in resilience should be considered and that the focus should not only be on maladaptive/passive/disengagement coping strategies but also on adaptive/active/engaged strategies to promote recovery and improve well-being [54,55]. In this regard, acceptance coping, a strategy not explored in this research, appears to be predictive of positive outcomes in the case of incurable cancer [56], whereas a lack of acceptance predicts poorer long-term psychological adjustment to breast cancer [57]. In a meta-analysis [51], the results of prospective studies about coping in breast cancer indicated that secondary-control coping (i.e., positive reappraisal, acceptance, and fighting spirit), aimed at facilitating the adaptation to stress without modifying the stressor or the related emotions, were associated with more positive effects and less negative ones. On the other hand, primary-control coping strategies (i.e., planning, social-support seeking, direct action), focusing on the modification of the stressor or related emotions, were associated with more positive effects but not with negative ones. As indicated by Cousson-Gélie (2019) [58] and Folkman (2008) [59], such results suggest a focus on positive and engagement coping strategies to promote positive effects and reduce negative ones.

Surprisingly, depression was predicted by any of the coping strategies, which is congruent with the results obtained by Barata et al. (2018) [29] showing that relationships between depression and coping disappear after a few weeks and that depression prior to hospitalization explains part of the symptomatology of depression after a few months. However, PTSD and anxiety are predicted by anxious preoccupations, which is not surprising given the high level of stress, and that maladaptive coping strategies tend to elicit a PTSD symptomatology [60]. Hopelessness-helplessness did not predict any of the psychological outcomes. However, such a coping strategy is related to a lack of control and self-efficacy, and a less optimistic view of the future prevents has a negative impact on the capacity for resilience, i.e., the ability to restore psychological functioning despite the stressful events, and this engenders higher levels of stress [54].

### 4.1. Study Limitations

The loss of part of our sample due to death and decreased desire to continue participating in this study between Time 1 (i.e., one week during transplantation) and Time 2 (i.e., five months after transplantation) necessarily limit the results of this study. Despite the heterogeneity of the medical centers selected to conduct this study, the study is not representative of the different bone marrow transplantation centers in France. For reasons of anonymity, it was not possible to know which center the patients came from. The results of the regression analyses may therefore be biased because our observations were not independent, which is the first assumption of a regression analysis. This type of study must also be replicated by combining psychological and medical variables, especially immunological variables (e.g., natural killer cells, lymphocytes, cytokines), which are rare in the case of HSCT [61]. For example, some studies reveal promising results concerning the effect of emotion-regulating coping on inflammatory biomarkers in the case of prostate cancer [62]. In addition, the mediation effect of coping between dispositional factors (e.g., dispositional mindfulness, acceptance, experiential avoidance, or optimism) and adjustment outcomes should be explored in further studies, given that some dispositions, such as experiential avoidance [63] or self-efficacy [58], seem to influence the process of coping and hence the subsequent psychological and physical health outcomes. Patients’ adjustment strategies were assessed at the precise moment of hospitalization and not several days or weeks after, which prevented an understanding of the effect of extended hospitalization and isolation. Change scores on coping strategies assessed at two different times would also be relevant for the testing of our prospective effects [1]. Only a fighting spirit, as an adaptive coping strategy, was explored in this study. However other secondary-control coping strategies identified by Kvillemo and Bränström (2014) [51], such as acceptance, should be explored as predictors of psychological health outcomes [24]. Such research would lead to the identification of the most appropriate engagement coping strategy to promote recovery and also target an appropriate intervention after HSCT, particularly with regard to the symptomatology of PTSD, which was particularly prevalent even five months after transplantation.

### 4.2. Clinical Implications

A focus on the transactional factors involved in HSCT recovery would allow for the introduction of relevant psychological preventive interventions. First, such results suggest that people should be provided with positive personal resources and adaptive coping strategies before hospitalization. This type of care appears relevant in that it will help patients better face this challenging hospitalization period and the next steps of the allograft, and help to improve psychological recovery after HSCT [64,65,66,67].

## 5. Conclusions

There are psychological considerations specific to HSCT, especially with regard to protective isolation and its consequences [68]. Exploring the factors which contribute to better mental and physical health at each step of the process, and their relationships, is fundamental in order to identify which interventions can be implemented in an effective way. However, the role of psychological factors and interventions on the physiological markers of HSCT success or fail rates are under-explored.

## Figures and Tables

**Figure 1 healthcare-11-01975-f001:**
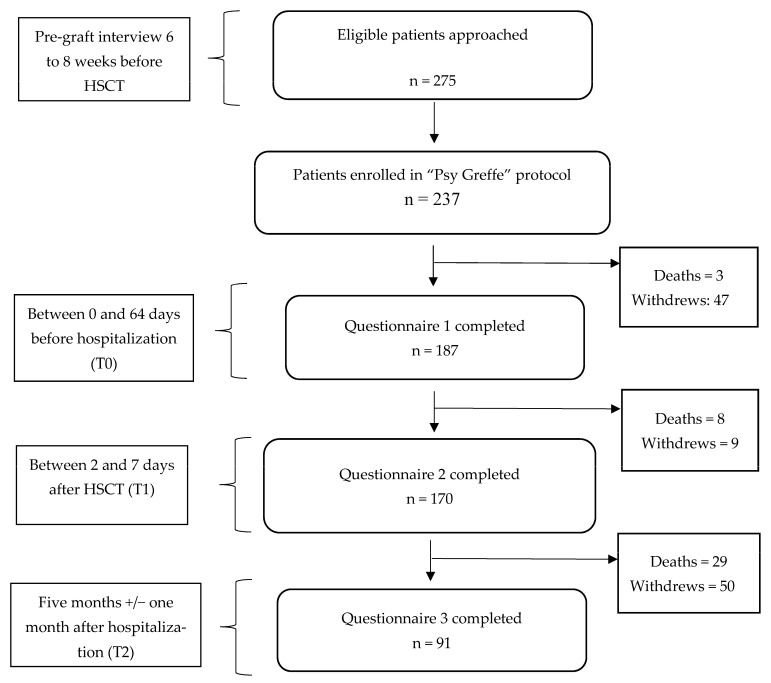
Flow diagram of protocol.

**Table 1 healthcare-11-01975-t001:** Descriptive Statistics for Socio Demographic and Medical Variables (*n* = 220).

	% (Excluding Missing Values)	Mean (*SD*)	*n*
Controlled socio demographic variables Age		52.03 (13.28)	217
Sex *(women)*	42.7		221
Marital Status *(married)*	46.4		181
Educational Level *(post-graduate)*	46.3		175
Socioprofessional Category *(employed)*	69.6		151
Follow-up (in months) Controlled medical variables		6.58 (4.04)	
Disease Status *Acute Leukemia* *Myelodysplastic Syndrome* *Myeloproliferative Neoplasia* *Non Hodgkin Lymphoma*	36 17.4 10.1 11.8		178
Alcohol consumption *(yes)* Smoking *(yes)* Physical Activity *(yes)* Body Mass Index Sleeping hours Number of transplantations Latency between disease diagnostic and transplantation (*in years*) Myeloablative conditioning Chronic GvHD Donor type *Identical sibling* *Mismatched unrelated* *Mismatched relative* *Matched unrelated* *Unrelated* *Matched other relative*	30.8 15.8 45.3 25.8 16.5 25.7 8.9 12.8 38 14 0.6	24.92 (4.61) 7.42 (1.15) 1.07 (0.3) 2.61 (4.41)	172 177 172 176 161 178 178 178 164 179
Dependant Medical Variables			
Latency engrafment (*in days*)		20.24 (6.95)	161
Acute GvHD Relapse Number of infections Death	51.5 14.8 16.4	2.14 (1.8)	171 162 170 177

Note: *n* = number of observations; *SD* = Standard Deviation.

**Table 2 healthcare-11-01975-t002:** Pearson’s Correlation Coefficients Between Perceived Stress, Perceived Social Support, Perceived Control and Adjustment Coping Strategies.

	SPS-R	PSS	IPC Internality	IPC Externality	FS	H/H	A	A/P	D
Social Provisions Scale-Revised (SPS-R)	-	−0.22 **	0.14	−0.19 *	−0.30 ***	−0.41 ***	−0.19 *	−0.28 ***	−0.23 **
Perceived Stress Scale (PSS)		-	0.01	0.34 ***	−0.46 ***	0.57 ***	0.04	0.60 ***	0.08
IPC internality			-	0.23 **	0.24 **	0.01	0.08	0.08	−0.02
IPC externality				-	−0.15	0.43 ***	0.29 ***	0.47 ***	0.27 ***
Fighting Spirit (FS)					-	−0.47 ***	0.10	−0.29 ***	0.04
Hopelessness/ Helplessness (H/H)						-	0.14	0.61 ***	0.21 *
Avoidance (A)							-	0.32 ***	0.46 ***
Anxious Preoccupations (A/P)								-	0.29 ***

Note: *** *p* < 0.001; ** *p* < 0.01; * *p* < 0.05.

**Table 3 healthcare-11-01975-t003:** Prospective Effects of Coping Strategies (at t1) on Mental Quality of Life, Mental Health, Post-Traumatic Stress Disorder (PTSD) and Resilience at Five Months Post-Transplantation (at t2).

	Fighting Spirit		Hopelessness/Helplessness		Avoidance		Anxious Preoccupations		Denial	
	Z-O	Adjusted *β*	Z-O	Adjusted *β*	Z-O	Adjusted *β*	Z-O	Adjusted *β*	Z-O	Adjusted *β*
Health Outcomes:										
Mental QoL ^a^	0.20	-	−0.35 **	0.01	−0.15	-	−0.52 ***	−0.42 **	−0.14	-
Anxiety ^b^	−0.27 *	−0.01	0.46 ***	−0.04	−0.02	-	0.62 ***	0.34 *	0.16	-
Depression ^c^	−0.30 *	0.10	0.34 **	0.10	−0.06	-	0.35 **	0.07	0.11	-
Happiness ^d^	0.53 ***	0.19 +	−0.45 ***	0.04	0.05	-	−0.47 ***	−0.17	0.07	-
PTSD ^a^	−0.06	-	0.30 *	−0.25 +	0.08	-	0.59 ***	0.72 ***	0.20	-
Resilience ^a^	0.40 ***	0.33 *	−0.36 **	−0.16	−0.08	-	−0.34 **	−0.05	−0.22	-

Note: The column Z-O depicts the zero-ordered effects of the variable, with other variables not included in the model. ^a^ Adjusted for Mental QoL at Time 0; ^b^ Adjusted for anxiety at Time 0 and occupation, ^c^ Adjusted for depression at Time 0; ^d^ Adjusted for happiness at Time 0 and sex. *** *p* < 0.001; ** *p* < 0.01; * *p* < 0.05; + *p* < 0.10.

## Data Availability

Dataset of this study is available on: https://doi.org/10.6084/m9.figshare.23266445 (accessed on 30 May 2023).

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
