# Peer review of "The Prospective Effects of Coping Strategies on Mental Health and Resilience at Five Months after HSCT"

_healthcare, 2023, doi:10.3390/healthcare11131975_

Round 1

Reviewer 1 Report (Previous Reviewer 3)

I believe a great job has been done to improve the manuscript and study has significant implications. 

Author Response

Dear reviewer,

Thank you for your comments about this paper.

Sincerely,

Maya Corman et al.

Reviewer 2 Report (Previous Reviewer 2)

Dear author,

Thank you for your efforts and the time spent on revising the first draft of your manuscript. However, after careful review, I must express my concern about the persistent methodological flaws present in the current version.

While I appreciate your efforts to make changes based on the initial feedback, I was disappointed to find that several of my significant methodological concerns were not adequately addressed. This left me with a sense that disregarded my initial comments and constructive suggestions, which were made with the aim to improve the quality and reliability of your study.

Given the substantial methodological issues that remain to be addressed, I found it challenging to proceed with the reviewing process beyond the results section. It's essential to understand that the peer-review process is a crucial aspect of scientific publication, facilitating the refinement and validation of a manuscript. Therefore, it's not productive to overlook substantive reviewer feedback.

In the future, I would strongly encourage you to carefully consider and fully address all feedback provided by reviewers. It shows respect for the reviewer's time and expertise, which, in turn, ultimately benefits the quality of your manuscript.

I sincerely hope that you can use these comments constructively to further refine your manuscript and improve the quality of your research.

Major remarks:

-          If you do not know from which medical center the data came from, you must add to the limitations that your residuals are not independent. In addition, the data within each medical center could be considered random, assuming proper random sampling procedures were followed. However, the combined data from all three centers might not be truly random or representative of the larger population due to differences in the populations served by each center and the stratified nature of the sampling. Therefore, your results cannot be generalized. This should be clearly stated in the manuscript or the manuscript will mislead the reader!

-          Table 3: I do not understand why you sometimes adjust for occupation, and sometimes for sex, and sometimes neither for occupation nor for sex? If I understand it correctly, you still have run multiple regressions with the same dependent variable, such as one regression for Fighting Spirit, including Mental Qol and Mental Qol at baseline, a 2nd regression for Fighting Spirit with Anxiety, Anxiety at baseline and Occupation, and so further. Am I right? If you have done this, you are still running multiple analyses on the same dependent variable, which leads to an accumulation of the alpha error. You should run a multiple regression for each dependent variable, or alternatively, adjust the results with the Bonferroni correction. I added a link to this topic in my last review. You cannot ignore my remarks, otherwise you will simply produce wrong results!

-          In the “Statistical Analysis” section, you wrote that you present z-scores, but you present regression coefficients and correlation coefficients, not z-scores. Please rewrite that you z-standardized the independent variables.

-          In Table 3, please state that you performed a multiple regression analysis. On page 8, line 584, you wrote that you did a correlation analysis and one sentence later you wrote that you did a multiple regression analysis. This is confusing.

-          Please include in the “Statistical Analysis” section how many multiple regression analyses you performed and what you used as the dependent variable and what you used as the independent variables. The paragraph on p. 8, line 585-594 should be added to the “Statistical Analysis” section since you are describing your methods. The Results section should only describe the result!

-          The testing of the assumptions should be mentioned in the Methods section, not in the Results section. Also, please double check which assumption should be tested for a linear regression. I do not understand why you tested for the normality distribution of depression, since only the residuals need to be normally distributed but not the independent variables.

-          Table 3: Why are there “pluses” added to some numbers?

-          Why are you comparing your sample to a randomized sample from the SF-12 validation study? Your sample is not randomized, because you do not know from which clinic your data came, so you cannot compare it to a randomized sample. You cannot generalize your findings.

Minor remarks:

-          The n in the flowchart should also not be capitalized.

-          The asterisks in the regression chart for the p-values do not make sense because you are reporting the change in z-values which are standardized values. Larger absolute values of the coefficients correspond to variables with greater influence on the dependent variable. This interpretation is valid because all variables are on the same scale. It is an old habit to compare the values of the coefficients with the number of asterisks. Please only use one asterisk for one significant value such as * p < 0.05 and not three significant values. There is no mathematical proof that a value with p <.001 fits better than a values with p < .05. You have z-standardized your independent variables to compare which independent variables have the biggest influence on the dependent variable (if you do a correct multiple regression analysis on the dependent variable).

I would recommend to have the manuscript edited by a native English speaker.

Author Response

Dear reviewer,

please find in attached file our responses to your comments. You hope this revised version of manuscript will be satisfying.

Sincerely,

Maya Corman et al.

Reviewer 3 Report (New Reviewer)

The manuscript describes the relationship between coping strategies and quality of life factors, social support, and perceived control.

The topic is relevant to psycho-oncology and healthcare, and the simultaneous consideration of different domains is a strength of the study.

The number of cases for the measurements at t0 and t1 is good, that for t2 (n=91) just sufficient.

The logic of the distribution of the questionnaires to the three time points is not well comprehensible, but the study was conducted in such a way that cannot be changed in retrospect. Thus, it is quite unusual that the effect of coping strategies (at t1) on resilience (at t2) is examined; resilience is normally considered to be more of an independent variable. This is a certain weakness of the study, which cannto be cahnged now.

The data collection instruments are largely well selected. However, when capturing the central construct, coping strategies, it is noticeable that some of the five dimensions present in the questionnaire are not coping dimensions inmy understanding: Fighting spirit concerns coping, but hopelessness, anxious obsession are actually not coping strategies, but dimensions of mental health.

Introduction: This gives a good insight into the topic.

line 76: “dimensions of individual who… “ What does this mean?

line 174: resiliency = > resilience

Methods: The methods are well described. I don't think the response alternatives to each questionnaire need to be described, but the descriptions are largely consistent in themselves and can be left that way.

The SF-12 should be described a little bit better. In the calculation of PCS and MCS all items are weighted; a calculation of Cronbach's alpha is not possible here.

For the happiness scale, please include the number of items.

The distribution of the questionnaires over the three time points is described in the description parts of the questionnaires, but for the understanding of the following analyses it would be helpful if it were tabulated again at which time points which questionnaires were used. Such an overview (probably in the form of a table) would be more useful than Figure 1.

Results:

Table 1 contains inconsistencies (at least in the presentation I downloaded). Why are age and gender data available from 217 subjects and 221 subjects, respectively? The number of cases at t1 is only 187.

The case number on disease status should be in the same row as Disease Status.

The four pairs of numbers (24.92 (4.61) to 2.61 (4.41)) are too low arranged; they surely refer to BMI use and so on. The numbers of cases in MBI etc. (172; 177;...) are obviously also wrong, probably too low in the arrangement. 

Line 502: If the range of the stress scale is from 14 to 26, it is implausible that the mean is 14.25; this is much too close to the lower limit.

Table 2, heading: Perason's Coefficients Correlations = > Correlation Coefficients .

It would be more readable if the left column entries were aligned with the left margin and not centered (all tables).

Point 3.2. deals with correlations between the questionnaires used at t1. It would be helpful for the reader to write this here so that one understands why exactly these variables are correlated with each other.

 I have doubts about the correlation of -.57 between stress and H/H. A/P and H/H are similar constructs, and they also correlate positively at 0.61. So they should also be about the same with stress. The correlation between stress and A/P is 0.60, but the correlation with H/H is the opposite (r=-0.57). Is this really true?

Table 3. In the text explaining the results of Table 3 we read that multiple regression analyses were performed. This sounds so that all independent variables are included simultaneously in the regression, but this is not the case. Please make clear in the text that the five independent variables were calculated in separate regression analyses.

Heading of Table 3: I recommend to also to clarify the time points : Effects of Coping Strategies (at t1) on Mental QoL… (at t2).

A table shold be understandable without the explaining text in the mansucript. From my point of view it would be better to change the colums and the rows, meaning that the dependent variables (Mental QoL…) should go from left to right, and that the independent variablens (coping strategies) are arranged at the left marging.

In addition, I would prefer also to present all (even the non-significant) adjusted beta coefficients. In the present form it is difficult to understand that missing coefficients are due to insignificant unadjusted betas.

The beta of 0.72 (even higher than the unadjusted score) is very surprising. In all other cases, the adjusted coefficients are smaller than the unadjusted ones. Maybe the adjustment by a third variable (mental QoL) is not appropriate here. Please recalculate and clarify.

line 618: Finally, the prospective effects of…. were also explored. Please write: the effects on what? (probably Mental qoL…., resilience)

I would prefer to see these analyses also in Table 3. If rows and columns are interchanged, it is easily possible to add these further analyses at the bottom of the table. Then one would also see the lack of significant associations in the table.

References: The references are good inprinciple; however, there is no reference newer than 2019.

The names of the Journal should be written in capitals, e.g., 4.:  Biology of Blood and Marrow Tranplantation, and so on.

Ref. 44: Publisher is missing

Ref 46: article number is mission

Ref 63 “October” is not necessary

The English is good.

Author Response

Dear reviewer,

please find in attached file responses to your comments. We hope this revised version of manuscript will be satisfying.

Sincerely,

Maya Corman et al.

Round 2

Reviewer 2 Report (Previous Reviewer 2)

Dear Authors,

I would like to extend my gratitude for the considerable effort you've put into revising your manuscript. However, I still have some comments and suggestions that I believe could further strengthen the manuscript. These are intended to refine and enhance the clarity of your work, ensuring its full potential is realised before final publication. I look forward to seeing the continued evolution of this valuable contribution to our field.

Statistical analyses:

·       Please perform a bootstrap analysis for depression if the normality assumption is violated.

·       Please report if there were any outliers or extreme values for the MLR.

·       Please describe if a linear relationship exists between dependent and predictor variables which is an assumption to do an MLR.

·       Please describe if the assumption of homoscedasticity has been met which is an assumption to do an MLR.

·       Please mention that your residuals are not independent (which is an assumption to do an MLR) because you cannot clearly distinguish the data between the 3 clinics. Weighing your data based on the distribution of demographic factors can help to adjust for potential confounding effects, but it cannot fully account for the lack of information about the clinics.

·       Please describe that you are comparing your sample to the SF-12 validation study at t2 and why you are doing so.

Results:

·       Please change “Results show that the score for 252 Quality of Life was significantly lower …” to “Results show that the score for Quality of Life was significantly lower in the sample…” p. 7, l. 252

·       Please change “Among the patients,…” to “Among the patients int the sample, …”, p. 7, l. 258

·       Please provide a scientific reference that there is a mathematical advantage of comparing different significance values. A citation guide is not a scientific reference. Here are some examples of the ongoing discussion about reporting significance levels

Heck, P., and Krueger, J., (2019) Putting the p-Value in Its Place

 Betensky, R., (2018) The p-Value Requires Context, Not a Threshold

Greenland, S., (2018) Valid p-Values Behave Exactly as They Should: Some Misleading Criticisms of p-Values and Their Resolution With s-Values

Johnson, V., (2019) Evidence From Marginally Significant t-Statistics

 Limitations:

·       Please add that the results of the regression analyses may be biased because your residuals are not independent.

Author Response

Dear reviewer,

Please find our responses about your comments. We hope this new version of the manuscript will satisfy your expectations.

Kind regards,

Maya Corman et al.

Reviewer 3 Report (New Reviewer)

The authors have adequately addressed most of the substantive points. However, the changes contain many errors.

The authors are inconsistent in writing p<0.01 or p<.01 ect. Please follow a consistent way throughout the manuscript. The same for r:  r = 0.34 or r = .34 ect.

The errors include:

l. 16: 170 Patients_(M…  => 170 patients (M

l. 32: “health in individual who beneficiates…”  incorrect expression

l. 70: show => shows

l. 76 ff: “They mainly reveal that greater the …..   such voidance… rarely they use….”:    incorrect language

l. 97 hematopoietic stem cell transplantation: this should be abbreviated

Figure 1: different fonts in the boxes.

Between 0 and 64 day before hospitalization (T0): This cannot be read correctly in the box

l. 139: come => came

l. 151: which unequely on voluntary basis …   incorrect

l. 223: analysis => analyses

l. 244 Socio Demographic

l. 151: Quality of Life => quality of life

l. 273: Correlations => Correlation

Table 3: the first two column pairs: Adjusted beta in bold; the remaining three colums not. Why?

. 321: p < 0.62. This is meaningless since the reader does not know: p<0.05? p<0.01? probably you want to express: p=0.62.

l. 473: incorrect citation

l. 566: https not necessary

The tables and the illustration extend over several pages. With better arrangement, this can be avoided. It would improve the readability significantly if the tables were designed without page breaks.

the corrections include several mistakes and inccrect expressions; see text

Author Response

Dear reviewer,

Please find our reponses to your comments. We hope this new version of the paper will satisfy your expectations.

Kinds regards,

Maya Corman et al.

This manuscript is a resubmission of an earlier submission. The following is a list of the peer review reports and author responses from that submission.

Round 1

Reviewer 1 Report

Weak and redundant sentences are utilized in several parts of the paper; rephrasing is required. 

Reviewer 2 Report

I think the topic of the presented manuscript is very interesting and a valuable research topic. However, I stopped the reviewing after the "Results", because there were already so many issues with the methods of the manuscript which have to be improved first.

Major Issues

Methods: Who was invited? How was the invited sample selected?

Statistical Analyses:

·       You wrote that you received questionnaires from 3 medical centers in France. Therefore, your sample is not random, but only random within each medical center. Please adjust your analyses by either doing a multilevel regression analysis or (if the differences between the medical centers are not large) by controlling for the 3 medical centers.

·       The statistical analyses should be described before reporting the results and not mixed in with the results. Please change this.

·       Are your analyses weighted or unweighted?

Results:

·       On page 6, l.202f, you wrote “At Time 2, the score for Quality of Life was significantly lower than the average score of the general population”. What statistical analysis did you do? I assume a t-test, but there is no table and no description of it. Please add this very important information.

·       Table 3: It looks like you did multiple regression analyses with the same outcome variables. This leads to an accumulation of the alpha error. You can correct this by using Bonferroni's correction or by doing a hierarchical regression analysis and adding your variable stepwise. The analyses as they are cannot be interpreted because of the accumulation of the alpha error.

·       I am very confused that there is no table for the logistic regression and the OR of “anxious preoccupations coping style was associated with an increased risk of developing PTSD five months after HSCT”  (p. 7). This confidence interval also looks awfully large. This means that some people in your sample had either a slightly higher risk of developing PTSD or a 33 times higher risk. This large confidence interval could be because you are ignoring the multilevel structure of your data. Since you did not describe your model, I cannot tell which other variables were included. You could try z-standardizing your predictor variables, especially if they differ greatly in their response categories, and see if the confidence interval becomes as large as it is right now.

Did you also test the assumptions of the regression analyses e.g. no outliers, linear releationship, independence of the residuals, etc.?

Minor Issues

Abstract: Please state the sample size

Methods:

·       Participants: It would be nice for the readers to have a flow chart of the number of participants, starting with the invitation to the study to the final sample size.

Tables:

·       Table 1: Please do not write a capital “N”. A capital “N” indicates the size of the population from which the sample is drawn. A lowercase “n” indicates the sample size. Please also report below the table that “n” is the number of observations and SD is the standard deviation. Tables should be readable at a first glance.

·       Table 2: Please add to the headline that the table presents results of a correlation analysis and please add that correlation coefficients are presented below the table. The stars on the correlation coefficients are misleading the reader into thinking that correlation coefficients with *** are better than the coefficients with *. The correlation should be interpreted by the size of the correlation coefficients, not by whether p was < 0.05 or <0.001. This is very arbitrary and there is no mathematical proof that a result with p <0.001 is better than a result with p < 0.05. I know that some papers report there finding this way, but just because it is a habit it does not mean one should repeat bad habits just for the habit itself. Decide for yourself if you want to use p < 0.001 or p < 0.05.

·       Table 3: The same which I wrote for Table 2 applies to Table 3. If you use standardized beta coefficients, they can also be interpreted by their size. If you had explained which beta coefficient you are presenting, I could have told, but you did not explain it properly. Please add this.

Reviewer 3 Report

The study is important and has good scope in developing psychological interventions for at-risk populations.  Some minor recommendations are made to improve the manuscript write-up.

Abstract

Report sample size in method of Abstract section.  When report Mean age also report S.D.. Also can report the figures by rounding off decimal points.

Introduction

It is well written, however, some alternative terms/synonyms used seems inappropriate. E.g. Line 76-78 need to be revised. It is not clear. The use of word ‘resignation’ seems not appropriate.

Method

Line 147. Rather than saying weak stress it's better to say low stress.

Please add Statistical Analysis applied and report about the assumptions assessed met for this analysis.

Results

Tables alignment in this file seems disrupted which misplaces the mean scores on Body Mass Index

Sleeping hours, Number of transplantations, Latency between disease diagnosis and transplantation (in

years).

Discussion

The discussion is well written and cover all aspects of findings.